# Pharmacogenomics of Vincristine-Induced Peripheral Neuropathy in Children with Cancer: A Systematic Review and Meta-Analysis

**DOI:** 10.3390/cancers14030612

**Published:** 2022-01-26

**Authors:** Aniek Uittenboogaard, Céline L. G. Neutel, Johannes C. F. Ket, Festus Njuguna, Alwin D. R. Huitema, Gertjan J. L. Kaspers, Mirjam E. van de Velde

**Affiliations:** 1Emma Children’s Hospital, Amsterdam UMC, Vrije Universiteit Amsterdam, Pediatric Oncology, 1105 AZ Amsterdam, The Netherlands; mi.vandevelde@amsterdamumc.nl; 2Princess Máxima Center for Pediatric Oncology, 3584 CS Utrecht, The Netherlands; A.D.R.Huitema-6@prinsesmaximacentrum.nl; 3Department of Neurosurgery, Radboud University Medical Center, 6525 GA Nijmegen, The Netherlands; celine.neutel@radboudumc.nl; 4Medical Library, Vrije Universiteit Amsterdam, 1081 HV Amsterdam, The Netherlands; h.ket@vu.nl; 5Department of Pediatric Oncology, Moi University, Eldoret 30107, Kenya; fnjuguna@ampath.or.ke; 6Department of Pharmacy & Pharmacology, The Netherlands Cancer Institute, 1066 CX Amsterdam, The Netherlands; 7Department of Clinical Pharmacy, University Medical Center Utrecht, Utrecht University, 3584 CX Utrecht, The Netherlands

**Keywords:** vincristine, vincristine-induced peripheral neuropathy, pediatric oncology, pharmacogenomics, meta-analysis, CYP3A5, children, CEP72, cancer

## Abstract

**Simple Summary:**

Vincristine is a drug that is part of the treatment for many children with cancer. Its main side-effect is vincristine-induced peripheral neuropathy (VIPN), which often presents as tingling, pain, and lack of strength in the hands and feet. It is not yet possible to predict which children will suffer from VIPN. In this review, we report on all genetic variations that are associated with VIPN. We found that variations in genes related to vincristine transport, cell structure, hereditary nerve disease, and genes without a previously known connection to vincristine or VIPN are related to VIPN. Variations in genes involved in vincristine breakdown are not significantly associated with VIPN. In conclusion, genetic variations affect a child’s tendency to develop VIPN. In the future, this information might be used to predict the risk of VIPN and adapt treatment on this.

**Abstract:**

Vincristine-induced peripheral neuropathy (VIPN) is a debilitating side-effect of vincristine. It remains a challenge to predict which patients will suffer from VIPN. Pharmacogenomics may explain an individuals’ susceptibility to side-effects. In this systematic review and meta-analysis, we describe the influence of pharmacogenomic parameters on the development of VIPN in children with cancer. PubMed, Embase and Web of Science were searched. In total, 1597 records were identified and 21 studies were included. A random-effects meta-analysis was performed for the influence of CYP3A5 expression on the development of VIPN. Single-nucleotide polymorphisms (SNPs) in transporter-, metabolism-, cytoskeleton-, and hereditary neuropathy-associated genes and SNPs in genes previously unrelated to vincristine or neuropathy were associated with VIPN. CYP3A5 expression status was not significantly associated with VIPN. The comparison and interpretation of the results of the included studies was limited due to heterogeneity in the study population, treatment protocol and assessment methods and definitions of VIPN. Independent replication is essential to validate the clinical significance of the reported associations. Future research should aim for prospective VIPN assessment in both a discovery and a replication cohort. Ultimately, the goal would be to screen patients upfront to determine optimal vincristine dosage with regards to efficacy and risk of VIPN.

## 1. Introduction

Vincristine is an important chemotherapeutic agent that is commonly used in treatment for pediatric cancers. It is approved by the United States Food and Drug Administration (FDA) for the treatment of acute lymphoblastic leukemia (ALL), Hodgkin and non-Hodgkin lymphoma, neuroblastoma, rhabdomyosarcoma, low-grade glioma and nephroblastoma. Furthermore, off-label uses include the treatment of Ewing sarcoma and medulloblastoma [1,2]. The main side-effect of vincristine is vincristine-induced peripheral neuropathy (VIPN), which often presents as a symmetric sensory-motoric neuropathy progressing distally to proximally [1,2]. Presenting signs include foot drop, loss of deep tendon reflexes, impaired balance, pain or tingling [1,2]. In addition, patients can suffer from autonomic symptoms such as constipation or orthostatic hypotension. The reported prevalence of VIPN varies, depending on assessment method and study population, but it is estimated that the majority of patients receiving vincristine will experience some form of VIPN during treatment [1,2,3,4]. Up to 30% of patients may suffer from severe VIPN, requiring dose reduction or cessation of treatment [3,5]. Suffering from VIPN is associated with a lower health-related quality of life, both by self- and proxy assessment and consistently when using different assessment tools for VIPN [6]. This effect of VIPN on health-related quality of life seems to persevere after treatment, as was shown in a recent study in ALL survivors in which over 16% suffered from long-term VIPN and experienced impact on both physical health and social functioning [7].

It is recognized that different populations might have an altered risk for VIPN [3]. Older age has been associated with an increased risk of VIPN, although results have been inconsistent [8,9,10,11,12]. In addition, white children appear to have a higher risk of VIPN than black children [3,9,12,13,14,15], which is corroborated by a recent study in Kenyan pediatric cancer patients in which only one out of 78 black patients developed severe VIPN and less than 5% developed clinically relevant VIPN, despite the use of sensitive assessment methods [16]. Interestingly, these children are being treated at a higher vincristine dose than what is common in Western countries (2.0 mg/m^2^ as opposed to 1.5 mg/m^2^) [1,16]. Studies assessing the relationship between VIPN and vincristine pharmacokinetics (PK) have shown inconsistent results. Some studies show a correlation between VIPN and PK parameters such as area under the curve (AUC) [17], an estimate of vincristine exposure, and intercompartmental clearance [18], whereas others do not confirm these findings [19,20,21,22]. Therefore, potential risk factors for VIPN could be genetic variations in genes involved in vincristine PK, such as variations in the cytochrome (CYP) 450 family of enzymes. Vincristine is predominantly metabolized by CYP3A4 and CYP3A5, of which the latter has a higher intrinsic clearance [23]. Genetic variants in both enzymes result in different metabolic activity [23,24]. Racial populations have different distributions of wild-type and variant CYP3A4/5 alleles [25,26,27]. Combined with the observation that black patients develop less VIPN, it has led to the hypothesis that faster clearance of vincristine in black children results in a lower risk of VIPN in comparison to white patients [14]. Indeed, several studies have described the effect of variations in CYP3A4 and CYP3A5 on the development of VIPN [8,13,14,16,20,28,29,30,31,32]. Differences in VIPN prevalence across populations may thus stem from variations in genetic background, which can be studied via the rapidly expanding field of pharmacogenomics.

Pharmacogenomics aims to assess the influence of genomics on an individuals’ treatment response and susceptibility to side-effects, such as VIPN [33,34]. Often, the effect of single nucleotide polymorphisms (SNPs) is assessed [35,36]. The frequency distribution of major and minor alleles varies across racial groups and study populations, which has been well characterized in large projects such as the 1000 Genomes Project and the genome Aggregation Database (gnomAD) [37,38]. Pharmacogenomics aims to find those SNPs or genetic variations that are biologically relevant [35,36]. Two main study designs have been used to assess this: candidate gene studies or population-based genome- or exome-wide association studies (GWAS or EWAS respectively) [39,40]. Candidate gene studies determine, a priori, a set of genes, based on available literature or mechanism of action, whose influence on a certain outcome is to be assessed [39]. Population-based GWAS or EWAS, on the other hand, assess the whole exome or genome (by whole exome sequencing (WES) or whole genome sequencing (WGS)) for genetic variation in relation to a certain outcome measure [39]. These studies may therefore result in previously unknown genotype—phenotype associations.

Pharmacogenomics can serve as a guidance tool for precision therapy in which a priori a patients’ genetic susceptibility for side-effects or therapeutic efficacy is determined. Although this has been implemented in clinical practice for some drugs, such as thiopurine methyltransferase (TPMT), this is currently not possible for vincristine [41,42]. Especially since there is a lack of understanding of what causes variability in VIPN across patients, pharmacogenomics can provide valuable insight into the pathogenesis of VIPN. If genes affecting vincristine PK are implicated, this may emphasize the potential of therapeutic drug monitoring. Moreover, since it is unlikely that VIPN is caused by differences in PK alone, variation in cellular sensitivity to vincristine and in neuronal pathways could be contributing factors. The implication of genes related to neuronal pathways, the cytoskeleton or cellular integrity with VIPN might then help guiding clinicians in deciding a priori if patients have a high chance of being developing (clinically relevant) VIPN and thus if patients should be monitored more closely than others, or even given an adapted vincristine dosage. In contrast, other patients might be identified who tolerate higher levels of vincristine and might thus not benefit from the generally applied dose capping at 1.5 mg/m^2^. Ultimately, the goal would be to develop a protocol for vincristine in which patients are stratified based on the presence of genetic polymorphisms and given a dosage that limits the risk of severe VIPN while maintaining the highest possible therapeutic efficacy. However, to explore this possibility, the first step is to provide a detailed overview of the effect of SNPs in all reported genes so far on VIPN. Therefore, in this systematic review, we aim to describe the influence of pharmacogenomic parameters on the development of VIPN in children with cancer. Furthermore, we performed a meta-analysis on the influence of CYP3A5 expression status on the development of VIPN.

## 2. Materials and Methods

### 2.1. Protocol and Registration

This study was performed according to the Preferred Reporting Items for Systematic Reviews and Meta-analyses guidelines (PRISMA) [43]. The study protocol was registered at the PROSPERO International prospective register of systematic reviews (registration number CRD42021210437) [44].

### 2.2. Eligibility Criteria

We included prospective and retrospective case-control or cohort studies assessing the relation between VIPN and pharmacogenomic parameters in five or more children with cancer. No restrictions regarding the characteristics of the children with cancer were applied. Pharmacogenomic parameters could include RNA variations such as microRNAs (miRNA) as well. To make sure we described the relationship between VIPN and pharmacogenomic parameters, we excluded descriptive reviews and studies in which no distinction could be made whether patients suffered from VIPN or neuropathy due to other causes, such as diabetes or Charcot-Marie-Tooth disease. If necessary, authors were contacted for clarification or additional data. Systematic reviews and meta-analyses were screened for additional inclusions.

### 2.3. Information Sources

PubMed, Embase, and Clarivate Analytics/Web of Science Core Collection were searched from inception up to 30 September 2021 (by AU and JCFK). Search terms were used as controlled vocabulary (e.g., MeSH) as well as free text terms for title, abstract and author keywords were amongst others: ‘pharmacogenetics’ and ‘child’ and ‘vincristine’ and ‘neuropathy’ or ‘constipation’. No limitations on language or date were applied. See Appendix A for the full search strategies in all databases.

### 2.4. Study Selection

Title and abstract screening were performed independently by two reviewers (AU and CLGN) based on pre-defined in- and exclusion criteria. Next, studies were screened full-text for eligibility by two reviewers (AU and CLGN). In case of disagreement, a third reviewer was consulted (MEvdV). Reasons for exclusion after full-text screening were documented. A meta-analysis was performed for studies assessing the relationship between CYP3A5 expression status and VIPN.

### 2.5. Risk of Bias Assessment

Risk of bias was assessed by two independent reviewers (AU and CLGN) according to a modified version of the quality assessment tool for quantitative studies by the Effective Public Health Practice Project (EPHPP), scoring each study as strong, moderate or weak in seven domains: study design, confounders, blinding, data collection methods, withdrawals and drop-outs, analysis, and selection of reported results [45]. The global rating of each study was determined as follows: no weak ratings of the domains resulted in an overall strong rating, one weak rating of the domains resulted in an overall moderate rating, and two or more weak ratings of the domains resulted in an overall weak rating.

### 2.6. Data Extraction and Synthesis

For the included studies, study and baseline characteristics were extracted according to a data extraction template (Appendix A). The main outcome was the relationship between VIPN and pharmacogenomic parameters expressed as an effect size (odds ratio (OR)) or *p*-value. ORs were either reported by studies or calculated by the authors. If studies reported multiple ORs using different definitions of cases of VIPN and controls, the significant result with highest clinical relevance was shown (severe VIPN or any grade VIPN). If studies performed both univariate and multivariate analyses, both effect sizes were included in this study.

For the meta-analysis on CYP3A5 expression status and VIPN, dominant OR was calculated if raw data were available, or pre-calculated ORs provided by the authors were used. CYP3A5 expressers were defined as having at least one functional allele (*1) and CYP3A5 non-expressers were defined as having only non-functional variant alleles. If needed, authors were contacted for missing data.

### 2.7. Statistical Analysis

The meta-analysis on CYP3A5 was performed in R, version 3.6.1, using the ‘meta’ package (Rstudio Inc.) [46]. A random-effects model was applied to pool odds ratios since considerable between-study heterogeneity was suspected. The Paule–Mandel procedure was used to estimate variance τ^2^. Heterogeneity was estimated using I^2^ with the following interpretations: I^2^ of 25% indicated low heterogeneity, I^2^ of 50% indicated moderate heterogeneity, and I^2^ of 75% indicated high heterogeneity. The inverse-variance approach was used with the ‘metagen’ command. If study heterogeneity I^2^ was higher than 50%, an assessment of outliers or influential cases was performed. The risk of publication bias was assessed via evaluation of a funnel plot and Egger’s testing for asymmetry. A two-sided *p*-value of <0.05 was considered statistically significant.

## 3. Results

### 3.1. Study Selection

We identified 1597 reports through database searching (Figure 1). After removal of duplicates, the abstracts and titles of 1367 records were screened. Of these, 109 were selected for full-text review. One report was sought for retrieval and no full-text version was published. Since insufficient data was available in the abstract, the report was excluded. Assessment of eligibility resulted in the exclusion of 88 reports based on: consisted of a narrative review (35 reports), no data available on pharmacogenomic parameters (24 reports), no data available on VIPN (13 reports), no administration of vincristine (12 reports), and same data were used as in another report (four reports). Finally, 21 reports were included in this systematic review.

### 3.2. Study Characteristics

Sociodemographic and baseline characteristics of the included studies can be found in Table 1. Eighteen studies followed a candidate gene approach [8,10,11,13,14,16,20,22,28,29,30,31,32,47,48,49,50,51], whereas three studies were population-based GWAS or EWAS [9,52,53] (Table 1). Four studies included a replication cohort to confirm their findings of the discovery cohort [8,9,52,53]. In total, the number of included patients with available genotype and VIPN data ranged from 24 to 1132. The majority of included patients were diagnosed with ALL and were white (Table 1). The prevalence of moderate to severe VIPN (grade 2–4) ranged from 19.5–53.2%, with the exception of the study by Skiles et al. who reported an incidence of 2.8% in black Kenyan patients [16]. Studies used different definitions of cases (patients with VIPN) and controls (patients without VIPN) (Table 2). Different measurement tools for VIPN were used in the different studies, most often the Common Terminology Criteria for Adverse Events (CTCAE), in which a subset of items was used to score peripheral neuropathy [8,9,10,13,14,16,20,29,30,47,50,51,52,53], followed by the modified Balis scale [9,16,31,32] and World Health Organization (WHO) scale [11,48,49] were used. Finally, The Children’s Cancer Group (CCG) toxicity criteria, National Cancer Institute (NCI) common toxicity criteria, Total Neuropathy Score—Pediatric Vincristine (TNS-PV) and pediatric modified total neuropathy score (ped-mTNS) were all used in one study each [16,22,28,53]. Seven studies assessed VIPN prospectively [9,16,20,28,32,47,53], while the rest of the studies assessed VIPN retrospectively.

### 3.3. Risk of Bias

An overview per domain of risk of bias can be found in Appendix A. Twelve [8,10,11,22,29,30,31,48,49,50,51,52] and nine studies [9,13,14,16,20,28,32,47,53] scored an overall strong and moderate rating on risk of bias, respectively. No studies received an overall weak rating on risk of bias.

**Table 1 cancers-14-00612-t001:** Sociodemographic and clinical characteristics of studies included in the systematic review.

Author and Year of Publication	Study Design	Patients with Genotype + VIPN Data (*n*)	Patient Characteristics				Vincristine Dosage		VIPN		Global Rating Risk of Bias Assessment
			Disease Studied	Age	Male (%)	Race (%)	Single Dosage, (per mg/m^2^ and max)	Cumulative Dosage (mg)	Method Used for VIPN Assessment	Prevalence VIPN	
Abaji et al., 2018—QcALL cohort [52]	EWAS	237	ALL	82.7% <10 y/o, 17.3% ≥10 y/o.	54.9	All white	1.5, max. 2.0	Not available	NCI-CTCAE 3.0, retrospective Grade 3–4 peripheral neuropathy	14.8%	Strong
Abaji et al., 2018—AIEOP cohort [52]	EWAS	405	ALL	83.2% <10 y/o, 16.8% ≥10 y/o.	53.1	All white	1.5, max. 2.0	Not available	NCI-CTCAE 3.0, retrospective Grade 3–4 peripheral neuropathy	3.2%	Strong
Abo–Bakr et al., 2017 [47]	Candidate gene	97	ALL	79.4% ≤10 y/o, 20.6% >10 y/o	58.8	All white	1.5, max. 2.0	Not available	NCI-CTCAE 3.0, prospective Foot drop, ileus, vocal cord paralysis, ptosis	Foot drop: 4.1%	Moderate
Aplenc et al., 2003 [28]	Candidate gene, case–control	533	ALL	70.0% ≤5 y/o, 30.0% >5 y/o	32.5	5.8 black 94.2 other	1.5, max. not available	46.5–64.5	CCG toxicity criteria, prospective Grade 3 or 4 peripheral neuropathy	5.3%	Moderate
Ceppi et al., 2014 [8]	Candidate gene	320	ALL	80.0% ≤10 y/o, 20.0% >10 y/o	55.3	All white	1.5–2.0, max. 2.0	73.5–74.0	NCI-CTCAE 3.0, retrospective Peripheral neuropathy	Grade 1–2: 20.0% Grade 3–4: 10.6%	Strong
Diouf et al., 2015—St. Jude cohort [9]	GWAS	St. Jude: 222.	ALL	68.9% ≤10 y/o, 31.1% >10 y/o	42.3	67.1 white, 19.8 black, 14.0 other	1.5, max. 2.0 COG: 1.5–	54.0	NCI-CTCAE 1.0, prospective Grade 2–4 peripheral neuropathy	28.8%	Moderate
Diouf et al., 2015—COG cohort [9]	GWAS	99	Relapsed ALL	47.5% ≤10 y/o, 52.5% >10 y/o	59.6	60.6 white, 1.0 black, 38.3 other	1.5–2.0, max. 2.0–2.5	78.0–97.5	Modified Balis scale, prospective Grade 2–4 peripheral neuropathy	22.9%	Moderate
Egbelakin et al., 2011 [29]	Candidate gene	107	ALL	Not available	Not available	92.5 white 0.9 black 6.5 other	1.5, max. 2.0	Not available	NCI-CTCAE 3.0, retrospective Peripheral and autonomic neuropathy	Grade 1–4: 98.1% Grade 3–4: 53.2%	Strong
Guilhaumou et al., 2011 [20]	Candidate–gene	24	Solid tumors	57.7% <10 y/o, 42.3% ≥10 y/o	57.7	All white	1.5, max 2.0	Mean (SD) at time of enrolment: 7.35 (5.30)	NCI-CTCAE 3.0, prospective Pain, peripheral neuropathy, gastro–intestinal toxicity	33.3%	Moderate
Gutierrez–Camino et al., 2016 [10]	Candidate gene	142	ALL	88.7% ≤10 y/o, 11.3% >10 y/o	57.0	All white	1.5, max 2.0	15.0–30.0	NCI-CTCAE 1.0, retrospective Grade 2–4 peripheral neuropathy	25.4%	Strong
Gutierrez–Camino et al., 2017 [48]	Candidate gene (miRNA)	155	ALL	Mean (SD): 5.1 (3.2) y/o	58.9	Mainly white	1.5, max 2.0	15.0–30.0	WHO criteria, retrospective Peripheral neuropathy	Grade 1–2: 16.0% Grade 3–4: 10.1%	Strong
Kayilioğlu et al., 2017 [30]	Candidate gene, case–control	Cases: 115 (VCR), controls: 50 (no VCR)	Cases: ALL and solid tumors. Controls: no neurological disorders or symptoms	Mean (SD): ALL 7.0 (4.6), solid tumors 7.5 (5.0), controls 10.2 (4.6)	ALL and solid tumors: 61.7 Controls: 62.0	All white	1.5, max 2.0	Mean (SD) total: ALL 7.71 (0.89), solid tumors 6.5 (1.5)	NCI-CTCAE 3.0, retrospective Grade 2–5 neurotoxicity	20.8%	Strong
Kishi et al., 2007 [13]	Candidate gene	240	ALL	70.4% ≤10 y/o, 29.6% >10 y/o	59.2	69.6 white 18.3 black 12.1 other	1.5, max 2.0	54.0–97.5	NCI-CTCAE 1.0, prospective/retrospective not available. Peripheral neuropathy and constipation	Grade 3: 12.1% Grade 4: 0.4%	Moderate
Li et al., 2019—POG cohort [53]	GWAS	1069.	ALL	Not available	52.3	All white	1.5, max not available	18–23 doses of 1.5 mg/m^2^	NCI-CTCAE 2.0, prospective Grade 3–5 peripheral neuropathy.	4.8%	Moderate
Li et al., 2019—ADVANCE cohort [53]	GWAS	63	ALL	Mean (SD): 8.2 (4.7) y/o	46.0	All white	1.5, max 2.0	Not available	TNS–PV, prospective. Sensory symptoms, temperature and vibration sensibility, strength, tendon reflexes.	Mean + SD: 3.8 (2.6)	Moderate
Lopez–Lopez et al., 2016 [11]	Candidate gene	133	ALL	Mean (SD): 5.5 (3.4) y/o	56.6	Mainly white	1.5, max 2.0	15.0–30.0	WHO criteria, retrospective Peripheral neuropathy	Grade 1–2: 18.4% Grade 3–4: 11.8%	Strong
Martin–Guerrero et al., 2019 [49]	Candidate gene	133	ALL	Mean (SD): 5.5 (3.4) y/o	56.6	Mainly white	1.5, max 2.0	15.0–30.0	WHO criteria, retrospective Grade 2–4 peripheral neuropathy	25.4%	Strong
McClain et al., 2018 [31]	Candidate gene	239	ALL	Mean (SD): 5.8 (3.9) y/o	53.1	All white	Not available	Mean (SD), at time of event: extensive metabolizers: 10.0 (5.7), intermediate: 13.4 (13.6), poor: 10.4 (8.9)	Modified Balis scale, retrospective Grade 3–4 peripheral neuropathy	Grade 3–4: 18.4%	Strong
Plasschaert et al., 2004 [22]	Candidate gene	52	ALL	73.1% < 10 y/o, 26.9% ≥ 10 y/o	61.5	98.1 white 1.9 other	Once 1.5, other doses 2.0, max. 2.5	13.5 mg/m^2^	NCI common toxicity criteria Constipation	Grade 1–2: 55.8%, Grade 3–4 26.9%	Strong
Renbarger et al., 2008 [14]	Race as surrogate for genotype, case–control	Cases: 21 black Controls: 92 white	ALL	Mean (SD): black: 8.2 (4.8) y/o, white: 5.0 (3.1) y/o	Cases + controls: 50.4	81.4 white 18.6 black	Not available	Mean (SD), Caucasians: 48.5 (14.3), AAs: 42.4 (11.6)	NCI-CTCAE 3.0, retrospective Neurotoxicity	Grade 1–4: 34.8% white, 4.8 black	Moderate
Sims et al., 2016 [32]	Candidate gene	52	BALL	77.4% < 10 y/o, 22.6% ≥ 10 y/o	62.2	68.5 white 31.5 black	1.5, max. 2.0	Not available	Modified Balis scale, prospective Peripheral neuropathy, constipation if grade 3–4	Grade 1–4: 80.6% white, 76.5% black	Moderate
Skiles et al., 2018 [16]	Candidate gene	72	Leukemia, lymphoma, solid tumors	Mean (SD): low expressers: 6.1 (5.2), intermediate: 6.5 (4.0), high: 6.1 (4.6)	53.8	All black Kenyan	2.0, max. 2.5	8.5 mg/m^2^	NCI-CTCAE 4.0, modified Balis scale, Faces Pain Scale, Pediatric Neuropathic Pain Scale, ped–mTNS, all prospective. Peripheral neuropathy and neuropathic pain	NCI–CTCAE: grade 2–4: 2.8%. Ped–mTNS: 4.3% 5 or higher.	Moderate
Wright et al., 2019 [51]	Candidate gene, case–control	Cases: 167 (VIPN), controls: 57 (no VIPN)	ALL	Median (IQR): cases 4.8 (3.3–9.0), controls: 5.4 (3.3–9.0)	Cases: 60.4, controls: 40.4	Mainly white	Not available	Median + IQR: cases: 61.4 (48.0–72.0), controls: 66.0 (51.0–74.8)	NCI-CTCAE 4.0, retrospective Peripheral neuropathy	Grade 2–4: 167 cases	Strong
Zgheib et al., 2018 [50]	Candidate gene	133	ALL	Mean (SD): 6.7 (5.0)	57.1	All white	Induction and re–induction: 1.5, max. 2.0. Continuation: 2.0, max. 2.0	Mean (SD), patients without VIPN: 66.0 (6.1), with VIPN grade 2–4: 27.9 (12.1)	NCI-CTCAE 4.0, retrospective Peripheral neuropathy	Grade 2–4: 19.5%	Strong

EWAS = exome-wide association study, ALL = acute lymphoblastic leukemia, NCI-CTCAE = National Cancer Institute—Common Toxicity Criteria for Adverse Events, CCG = Children’s Cancer Group, GWAS = genome-wide association study, SD = standard deviation, miRNA = microRNA, WHO = World Health Organization, TNS-PV = Total Neuropathy Score—Pediatric Vincristine, ped-mTNS = pediatric modified total neuropathy score, NCI = National Cancer Institute, IQR = interquartile range.

**Table 2 cancers-14-00612-t002:** Single-nucleotide polymorphisms that were significantly associated with vincristine-induced peripheral neuropathy in the pediatric oncology population.

Gene	SNP	Allele, Major/Minor	Author and Year of Publication	MAF (%)	Number of Patients (*n*)		Method Effect Size	Effect Size with 95% CI (If Applicable)	Effect
					Cases of VIPN *	Controls *			
Transport									
ABCB1	rs4728709	C/T	Ceppi et al., 2014 [8]	TT/TC: 17.1 CC: 82.9	63 (grade 1–2)	214 (grade 0)	Dominant OR	0.3 (0.1–0.9)	Protective ^1^
	rs10244266	T/G	Lopez-Lopez et al., 2016 [11]	14.3	46 (WHO grade 1–4)	103 (WHO grade 0)	Dominant OR	2.60 (1.16–5.83)	Risk ^2^
	rs10268314	T/C	Lopez-Lopez et al., 2016 [11]	14.3	27 (WHO grade 1–2)	103 (WHO grade 0)	Dominant OR	3.19 (1.23–8.25)	Risk ^2^
	rs10274587	G/A	Lopez-Lopez et al., 2016 [11]	14.6	27 (WHO grade 1–2)	103 (WHO grade 0)	Dominant OR	3.48 (1.36–8.86)	Risk ^2^
ABCC1	rs1967120	T/C	Lopez-Lopez et al., 2016 [11]	27.3	18 (WHO grade 3–4)	103 (WHO grade 0)	Dominant OR	0.29 (0.09–0.99)	Protective ^2^
	rs3743527	C/T	Lopez-Lopez et al., 2016 [11]	19.7	46 (WHO grade 1–4)	103 (WHO grade 0)	Dominant OR	0.32 (0.13–0.79)	Protective ^2^
	rs3784867	C/T	Wright et al., 2019 [51]	32.0	170 (grade 2–4)	57 (grade 0)	Additive OR	4.91 (1.99–12.10)	Risk ^3^
	rs11642957	T/C	Lopez-Lopez et al., 2016 [11]	48.1	46 (WHO grade 1–4)	103 (WHO grade 0)	Dominant OR	0.43 (0.19–0.98)	Protective ^2^
	rs11864374	G/A	Lopez-Lopez et al., 2016 [11]	24.4	46 (WHO grade 1–4)	103 (WHO grade 0)	Dominant OR	0.35 (0.15–0.79)	Protective ^2^
	rs12923345	T/C	Lopez-Lopez et al., 2016 [11]	15.4	46 (WHO grade 1–4)	103 (WHO grade 0)	Dominant OR	2.39 (1.08–5.25)	Risk ^2^
	rs17501331	A/G	Lopez-Lopez et al., 2016 [11]	13.2	46 (WHO grade 1–4)	103 (WHO grade 0)	Dominant OR	2.50 (1.10–5.68)	Risk ^2^
ABCC2	rs12826	G/A	Lopez-Lopez et al., 2016 [11]	42.6	46 (WHO grade 1–4)	103 (WHO grade 0)	Dominant OR	0.24 (0.10–0.54)	Protective
	rs3740066	G/A	Lopez-Lopez et al., 2016 [11]	36.2	46 (WHO grade 1–4)	103 (WHO grade 0)	Dominant OR	0.23 (0.10–0.53)	Protective
	rs2073337	A/G	Lopez-Lopez et al., 2016 [11]	45.8	18 (WHO grade 3–4)	103 (WHO grade 0)	Dominant OR	0.35 (0.10–1.24)	Protective
	rs4148396	C/T	Lopez-Lopez et al., 2016 [11]	42.1	46 (WHO grade 1–4)	103 (WHO grade 0)	Dominant OR	0.36 (0.16–0.81)	Protective
	rs11190298	G/A	Lopez-Lopez et al., 2016 [11]	45.0	46 (WHO grade 1–4)	103 (WHO grade 0)	Recessive OR	2.44 (1.01–5.86)	Risk
ABCC1/RALPB1: miR–3117	rs12402181	G/A	Gutierrez–Camino et al., 2017 [48]	14.8	19 (WHO grade 3–4)	128 (WHO grade 0)	Dominant OR	0.13 (0.02–0.99)	Protective ^2^
Vincristine metabolism									
CYP3A4	rs2740574	A/G(*1B)	Aplenc et al., 2003 [28]	8.6	28 (CCG grade 3–4)	505 (CCG grade 0–2)	Allelic OR	0 (0–0.75)	Protective ^2^
			Guilhaumou et al., 2011 [20]	6.3	Nr of neurotoxicity events		Chi–square	*p* = 1.00	Not significant
			Kishi et al., 2007 [13]	AA: 79.6 AG + GG: 20.4	30 (grade 2–4)	210 (grade 0–1)	Dominant OR	1.37 (0.57–3.29)	Not significant
GSTM1	Deletion	Non–null/null	Kishi et al., 2007 [13]	Non–null: 57.5 Null: 42.5	30 (grade 2–4)	210 (grade 0–1)	OR	0.46 (0.22–0.94)	Protective^2^
VDR	rs1544410	G/A	Kishi et al., 2007 [13]	GG: 45.8 AA and AG: 54.2	30 (grade 2–4)	210 (grade 0–1)	Recessive OR	2.22 (1.06–4.67)	Risk
Cytoskeleton–associated									
ACTG1	rs1135989	G/A	Ceppi et al., 2014 [8]	36.5	38 (grade 3–4)	214 (grade 0)	Dominant OR	2.8 (1.3–6.3)	Risk ^1^
CAPG	rs2229668	G/A	Ceppi et la. 2014 [8]	12.6	39 (grade 3–4)	214 (grade 0)	Dominant OR	2.1 (1.1–3.7)	Risk ^1^
	rs3770102	C/A	Ceppi et al., 2014 [8]	41.4	39 (grade 3–4)	214 (grade 0)	Dominant OR	0.1 (0.01–0.8)	Protective ^1^
CEP72	rs924607	C/T	Diouf et al., 2015—St. Jude cohort [9]	36.7	64 (grade 2–4)	158 (grade 0)	Recessive OR	5.5 (2.5–12.2)	Risk
			Diouf et al., 2015—COG cohort [9]	36.4	22 (grade 2–4)	74 (grade 0)	Recessive OR	3.8 (1.3–11.4)	Risk
			Gutierrez–Camino et al., 2016 [10]	39.4	36 (WHO grade 2–4)	106 (WHO grade 0–1)	Recessive OR	0.7 (0.2–2.4)	Not significant
			Wright et al., 2019 [51]	TT: 13.5 CT and CC: 86.5	156 (grade 2–4)	56 (grade 0)	Recessive OR	3.4 (0.9–12.6)	Not significant
			Zgheib et al., 2018 [50]	36.9	23 (grade 2–4)	107 (grade 0–1)	Recessive OR	1.04 (0.32–3.43)	Not significant
MAPT	rs11867549	A/G	Martin–Guerrero et al., 2019 [49]	22.5	18 (WHO grade 3–4)	103 (WHO grade 0)	Dominant OR	0.21 (0.04–0.96)	Protective ^2^
SYNE2	rs2781377	G/A	Abaji et al., 2018—QcALL cohort [52]	7.8	35 (grade 3–4)	201 (grade 0)	Additive OR	2.5 (1.2–5.2)	Risk
TUBB2B: miR–202	rs12355840	T/C	Martin–Guerrero et al., 2019 [49]	23.4	27 (WHO grade 1–2)	103 (WHO grade 0)	Dominant OR	2.88 (1.07–7.72)	Risk
Hereditary neuropathy									
SLC5A7	rs1013940	T/C	Wright et al., 2019 [51]	15.2	170 (grade 2–4)	57 (grade 0)	Additive OR	8.60 (1.68–44.15)	Risk ^3^
Other (GWAS/EWAS studies)									
BAHD1	rs3803357	C/A	Abaji et al., 2018—QcALL cohort [52]	41.7	35 (grade 3–4)	201 (grade 0)	Dominant OR	0.35 (0.2–0.7)	Protective
COCH	rs1045466	T/G	Li et al., 2020—POG cohort [53]	38	Maximum neuropathy score		Dominant HR	0.27 (0.16–0.50)	Protective
			Li et al., 2020—ADVANCE cohort [53]	33			Linear regression	−3.56 (−5.45;−1.67)	Protective
Chromosome 12/ chemerin	rs7963521	T/C	Li et al., 2020—POG cohort [53]	41	Maximum neuropathy score		Additive HR	2.23 (1.49–3.35)	Risk
			Li et al., 2020—ADVANCE cohort [53]	43			Additive HR	2.16 (0.53–3.70)	Not significant
ETAA1	rs17032980	A/G	Diouf et al., 2015—St. Jude cohort [9]	26.6	64 (grade 2–4)	158 (grade 0)	Allelic OR	3.17 (1.95–5.17)	Risk
			Diouf et al., 2015—COG cohort [9]	19.2	22 (grade 2–4)	74 (grade 0)	Allelic OR	10.4 (2.97–36.15)	Risk
MRPL4	rs10513762	C/T	Abaji et al., 2018—QcALL cohort [52]	7.0	35 (grade 3–4)	202 (grade 0)	Dominant OR	3.3 (1.4–7.7)	Risk
MTNR1B	rs12786200	C/T	Diouf et al., 2015—St. Jude cohort [9]	22.7	64 (grade 2–4)	158 (grade 0)	Allelic OR	0.23 (0.13–0.40)	Protective
			Diouf et al., 2015—COG cohort [9]	20.7	22 (grade 2–4)	74 (grade 0)	Allelic OR	0.24 (0.08–0.76)	Protective
			Zgheib et al., 2018 [50]	18.1	23 (grade 2–4)	107 (grade 0–1)	Dominant OR	0.59 (0.22–1.62)	Not significant
NDUFAF6	rs7818688	C/A	Diouf et al., 2015—St. Jude cohort [9]	12.6	64 (grade 2–4)	158 (grade 0)	Allelic OR	4.26 (2.45–7.42)	Risk
			Diouf et al., 2015—COG cohort [9]	14.1	22 (grade 2–4)	74 (grade 0)	Allelic OR	4.59 (1.35–15.59)	Risk
TMEM215	rs4463516	C/G	Diouf et al., 2015—St. Jude cohort [9]	33.6	64 (grade 2–4)	158 (grade 0)	Allelic OR	3.17 (1.95–5.17)	Risk
			Diouf et al., 2015—COG cohort [9]	24.2	22 (grade 2–4)	74 (grade 0)	Allelic OR	4.94 (1.65–14.79)	Risk
miRNA									
miR–4481	rs7896283	T/C	Gutierrez–Camino et al., 2017 [48]	37.5	19 (WHO grade 3–4)	128 (WHO grade 0)	Dominant OR	4.69 (1.43–15.43)	Risk ^2^
miR–6076	rs35650931	G/C	Gutierrez–Camino et al., 2017 [48]	8.7	47 (WHO grade 1–4)	128 (WHO grade 0)	Dominant OR	0.22 (0.05–0.97)	Protective ^2^

SNP = single nucleotide polymorphism, MAF = minor allele frequency, CI = confidence interval, OR = odds ratio, ABCB1 = ATP binding cassette subfamily B member 1, ABCC1 = ATP binding cassette subfamily C member 1, ABCC2 = ATP binding cassette subfamily C member 2, RALPB1 = RalA binding protein 1, miR = microRNA, CYP3A4 = cytochrome P450 3A4, GSTM1 = glutathione S-transferase mu 1, VDR = vitamin D receptor, CAPG = capping actin protein gelsolin like, CEP72 = centrosomal protein 72, MAPT = microtubule associated protein tau, TUBB2B = tubulin beta 2B class IIB, ACTG1 = actin gamma 1, SYNE2 = spectrin repeat containing nuclear envelope protein 2, SLC5A7 = solute carrier family 5 member 7, BAHD1 = bromo adjacent homology domain containing 1, COCH = cochlin, ETAA1 = Ewing’s tumor-associated antigen 1, MRPL4 = mitochondrial ribosomal protein L4, MTNR1B = melatonin receptor 1B, NDUFAF6 = NADH: ubiquinone oxidoreductase complex assembly factor 6, TMEM215 = transmembrane protein 215. * Grades are referring to CTCAE grades unless mentioned otherwise. ^1^ Significance threshold not adjusted for multiple comparisons. ^2^ Significance threshold was not met after correcting for multiple comparisons. ^3^ Significance threshold was not adjusted for multiple comparisons, but associations *p* < 0.001 were prioritized. Odds ratios (OR) were defined as following: recessive OR meant that the risk of VIPN increased y-fold if two copies of the minor allele (genotype: aa) or genetic variation were present; dominant OR meant that the risk of VIPN increased y-fold if either one or two copies of the minor allele were present (genotypes: Aa or aa); allelic OR meant that the risk of VIPN increased y-fold with each additional copy of the minor allele or genetic variation; and the additive OR meant that the risk of VIPN increased y-fold for the heterozygous genotype (Aa) and 2y-fold for the homozygous variant genotype (aa).

### 3.4. Association between Pharmacogenomic Parameters and VIPN

Table 2 and Table 3 show an overview of all SNPs found to have a statistically significant and non-significant association with VIPN, respectively. Figure 2 shows a schematic overview of the function of genes associated with VIPN. Sixteen SNPs in three ATP-binding cassette transporter genes (ABCB1, ABCC1, ABCC2) and one SNP in an miRNA targeting ABCC1/RalA binding protein 1 (RALPB1) were described to be significantly associated with VIPN (Table 2). Ten SNPs were associated with a protective effect against VIPN, whereas seven SNPs were associated with an increased risk of VIPN. Of note, the strongest protective associations with high precision were reported for SNPs rs3740066 and rs12826 in ABCC2 (OR 0.23, 95% CI 0.10−0.53, and 0.24, 95% CI 0.10−0.54 respectively). The strongest risk association with acceptable precision was reported for rs3784867 in ABCC1 (OR 4.91, 95% CI 1.99−12.10).

In terms of metabolism-associated genes, a deletion in glutathione S-transferase mu 1 (GSTM1) and an SNP in vitamin D receptor (VDR) were implicated with a heightened and a decreased risk to VIPN, respectively (Table 2) [13]. Furthermore, six SNPs in cytoskeleton-associated genes or in miRNAs targeting those were associated with VIPN (microtubule associated protein tau (MAPT), targeting tubulin beta 2B class IIB (TUBB2), actin gamma 1 (ACTG1), capping actin protein gelsolin like (CAPG) and spectrin repeat containing nuclear envelope protein 2 (SYNE2)) (Table 2). Of those, two SNPs were related to microtubules (MAPT and TUBB2) and associated with a protective effect and an increased risk of VIPN, respectively (Table 2) [49]. The four other SNPs were located in cytoskeleton-associated genes (ACTG1, CAPG, and SYNE2) and associated with a CTCAE grade 3−4 VIPN (Table 2) [8,52]. The latter passed the stringent significance threshold for multiple comparisons, but the results could not be confirmed in a replication cohort [52]. The strongest protective association was noted for SNP rs3770102 in CAPG with an effect size of 0.1, although the uncertainty was high (95% CI 0.01−0.8). One SNP in a gene associated with hereditary neuropathies (solute carrier family 5 member 7 (SLC5A7)) resulted in an increased susceptibility to VIPN (Table 2) [51]. The reported effect size was large, but the size of the confidence interval indicated relatively high uncertainty (OR 8.60, 95% CI 1.68−44.15) Except for the SNP in SYNE2, all aforementioned SNPS were solely assessed in a discovery cohort and no replication studies were performed for any of those associations [52].

Four studies assessed the influence of SNP rs924607 in CEP72 on the development of CTCAE or WHO grade 2−4 VIPN [9,10,50,51]. In both their discovery and replication cohort, Diouf et al. described an increased risk of VIPN in patients with the risk genotype [9]. The strongest association was seen in the discovery cohort (OR 5.5, 95% CI 2.5−12.2); this effect size was smaller in the replication cohort (OR 3.8, 95% CI 1.3−11.4) where the prevalence of VIPN was also slightly lower (28.8 and 22.9% respectively). Three replication studies could not confirm these findings [10,50,51].

GWAS or EWAS demonstrated significant associations between VIPN and eight SNPs in genes previously not associated with neuropathy, vincristine mechanism of action or metabolism (Table 2). All studies first reporting these associations made use of both a discovery and replication cohort to validate their results [9,52,53]. SNPs in cochlin (COCH), Ewing’s tumor-associated antigen 1 (ETAA1), melatonin receptor 1B (MTNR1B), NADH: ubiquinone oxidoreductase complex assembly factor (NDUFAF6), and transmembrane protein 215 (TMEM215) were significantly associated with VIPN both in a discovery and replication cohort, whereas this relationship was only established in the discovery cohort for SNPs in bromo adjacent homology domain containing 1 (BAHD1), chromosome 12/chemerin, and mitochondrial ribosomal protein L4 (MRPL4). The described SNPS in BAHD1 and COCH were protective against VIPN. The strongest protective association with high precision was reported for the latter (OR 0.27, 95% CI 0.16−0.50). The SNPs in chromosome 12/chemerin, ETAA1, MRPL4, NDUFAF6, and TMEM215 were associated with an increased risk of VIPN. The SNP in ETAA1 showed a strong effect on risk of VIPN, especially in the replication cohort of Diouf et al., although the precision was relatively low (OR 10.4, 95% CI 2.97−36.15). Moreover, the SNPs in NDUFAF6 and TMEM215 also showed relatively large effect sizes with acceptable uncertainty in both a discovery and replication cohort. Finally, Diouf et al. described an SNP in melatonin receptor 1B (MTNR1B) as protective against VIPN both in a discovery and replication cohort with a large effect size and high precision (OR 0.23, 95% CI 0.13−0.40, and OR 0.24, 95% CI 0.08−0.76), but another study by Zgheib et al. could not confirm these results [9,50]. All significant associations passed the stringent threshold for multiple comparisons.

Gutierrez-Camino et al. found two SNPs in miRNA to be associated with VIPN, of which one miRNA could be related to the axon-guidance pathway, whereas the other could not be related to any known vincristine- or neurotoxicity-related pathway (Table 2) [48].

Several studies assessed the influence of covariates such as cumulative vincristine dosage, treatment protocol, and patient characteristics on their results, but these covariates did not have a significant influence on the reported associations (Appendix A). Only the significant associations reported by Diouf et al. did not maintain their significance when corrected for genetically defined ancestry and cumulative vincristine dosage (Appendix A) [9].

**Table 3 cancers-14-00612-t003:** Single-nucleotide polymorphisms that were not significantly associated with vincristine-induced peripheral neuropathy in the pediatric oncology population.

Gene	SNP	Author and Year of Publication
ABCB1	rs1045642	Plasschaert et al., 2004 [22], Ceppi et al., 2014 [8], Zgheib et al., 2018 [50]
	rs1128503	Ceppi et al., 2014 [8], Zgheib et al., 2018 [50]
	rs2032582	Plasschaert et al., 2004 [22], Ceppi et al., 2014 [8]
ABCC2	rs717620	Zgheib et al., 2018 [50]
ACTG1	rs1139405	Ceppi et al., 2014 [8]
	rs7406609	Ceppi et al., 2014 [8]
CAPG	rs6886	Ceppi et al., 2014 [8]
CYP1A1	rs4646903	Abo-Bakr et al., 2017 ^1^ [47]
GSTP1	rs1695	Kishi et al., 2007 [13], Abo-Bakr et al., 2017 ^1^ [47]
GSTT1	Deletion	Kishi et al., 2007 [13]
MAP4	rs11268924	Ceppi et al., 2014 [8]
	rs1137524	Ceppi et al., 2014 [8]
	rs1875103	Ceppi et al., 2014 [8]
	rs11711953	Ceppi et al., 2014 [8]
MDR1	Exon 21, G > T/A	Kishi et al., 2007 [13]
	Exon 26, C/T	Kishi et al., 2007 [13]
MTHFR	rs1801133	Kishi et al., 2007 [13]
	rs1801131	Kishi et al., 2007 [13]
SLC19A1	rs1051266	Kishi et al., 2007 [13]
TPMT	Combined genotypes: 238GG, 460GG, 719AA/others	Kishi et al., 2007 [13]
TUBB	rs6070697	Ceppi et al., 2014 [8]
	rs10485828	Ceppi et al., 2014 [8]
TYMS	Enhancer repeat: others/3AND3	Kishi et al., 2007 [13]
UGT1A1	Enhancer repeat: others/7AND7	Kishi et al., 2007 [13]
VDR	rs2228570	Kishi et al., 2007 [13]
XRCC1	rs1799782	Abo-Bakr et al., 2017 ^1^ [47]

CYP1A1 = cytochrome P450 family 1 subfamily A member 1, GSTP1 = glutathione S-transferase pi 1, GSTT1 = glutathione S-transferase theta 1, MAP4 = microtubule-associated protein 4, MDR1 = multidrug resistance mutation 1, MTHFR = methylenetetrahydrofolate reductase, SLC19A1 = solute carrier family 19 member 1, TPMT = thiopurine methyltransferase, TYMS = thymidylate synthetase, UGT1A1 = uridine glucuronosyltransferase 1A1, XRCC1 = X-ray repair cross-complementing protein 1. ^1^ Association could not be tested due to small number of patients with VIPN.

### 3.5. CYP3A4 and CYP3A5

In regard to CYP3A4, Aplenc et al. found an SNP in CYP3A4 to be protective against VIPN [28], but two follow-up studies could not replicate these findings (Table 2) [13,20]. Furthermore, ten studies assessed the influence of CYP3A5 expression on the development of VIPN [8,13,14,16,20,28,29,30,31,32]. Of those studies, nine either presented a pre-calculated OR or raw data to calculate an OR and could thus be included in the meta-analysis [8,13,14,16,20,28,29,31,32]. If possible, dominant ORs were calculated based on data presented in the article or additional data provided by the authors (Appendix A). As shown in Figure 3, there was no statistically significant pooled effect between CYP3A5 expression status and the development of VIPN (pooled OR 0.69, 95% CI 0.38−1.26, I^2^ = 50%, τ^2^ = 0.33). The study by Kayilioğlu et al., which could not be included in the meta-analysis due to the unavailability of appropriate data, did not find a significant association either (Appendix A). However, all studies either found no effect of CYP3A5 status or found that expression of CYP3A5 was a protective factor for VIPN; the opposite was not reported. Of note, the included studies all used different definitions for cases (patients with VIPN) and controls (patients without VIPN) (Appendix A), likely contributing to the moderate heterogeneity. In addition, all studies performed genotyping to determine CYP3A5 expression status, whereas Renbarger et al., the study with the strongest association, used race as a surrogate. All studies compared expressers of CYP3A5 to non-expressers, except for Ceppi et al. who calculated allelic OR (Appendix A). Evaluation of the funnel plot and Egger’s test for asymmetry were not indicative of obvious publication bias (*p*-value Egger’s test = 0.40) (Appendix A).

## 4. Discussion

This systematic review shows that pharmacogenomic parameters have a significant influence on VIPN in children with cancer and show potential for clinical relevance. Several SNPs in genes related to vincristine metabolism, hereditary neuropathy, the cytoskeleton and microtubules have been associated with VIPN. Furthermore, population-based GWAS and EWAS identified significant interactions with SNPs in genes previously unrelated to VIPN or vincristine. Our meta-analysis showed that CYP3A5 expression status was not a significant risk factor for VIPN.

Several significant associations were found between SNPs in the ABC family of genes (ABCB1, ABCC1, ABCC2). These genes code for transmembrane proteins that mediate vincristine efflux across cell membranes; variations may thus contribute to different vincristine levels and therefore VIPN (Figure 2) [54,55]. Three candidate gene studies described associations between ABCB1, ABCC1 and ABCC2 and VIPN, all in children with ALL [8,11,51]. Of note, the vast majority of the associations were reported in the same discovery cohort (Lopez-Lopez et al.) [11]. Interestingly, all associations between SNPs in ABCC2 and VIPN passed a significance threshold corrected for multiple comparisons and showed the largest effect sizes, suggesting that a stronger relationship may exist between ABCC2 and VIPN than between ABCB1/ABCC1 and VIPN. Indeed, several cell line studies have shown that ABCC2 function was associated with vincristine resistance or sensitivity [56,57,58]. However, none of the reported associations have been replicated in other cohorts and results should thus be interpreted with caution.

Furthermore, SNPs in cytoskeleton-associated genes were associated with VIPN (Figure 2). Vincristine exerts its cytostatic effect via binding to the β-subunit of tubulins, which inhibits microtubule polymerization and consequently causes arrest of mitosis in the metaphase [1,59]. During cell division, there is a well-known interaction between microtubules and the actin cytoskeleton; the latter contributes to mitotic spindle assembly and formation [60,61,62]. It is possible that SNPs in genes that affect microtubule formation or the actin cytoskeleton affect binding of vincristine to tubulins or the effect of vincristine binding to tubulins. While this can result in an altered risk of VIPN, one could also hypothesize that this influences the effect of vincristine on mitotic spindle disintegration and thus ultimately the cytotoxic effect. Should that be the case, patients with a lower risk of VIPN might also experience less antitumor effect in comparison with patients with a higher risk of VIPN, which would argue for dose individualization in which standard dose capping is not applied to every patient. Future studies assessing the relationship between VIPN incidence and long-term treatment outcome, correcting for received cumulative vincristine dosage, may provide further insight. Of note, the studies reporting these associations concerned predominantly white patients with ALL and except for one study, the reported associations have not been assessed in a replication cohort [8,49,52]. Therefore, these results regarding SNPs in microtubule- and cytoskeleton-associated genes should be interpreted with caution until independent replication is performed. An association that has been replicated in several independent studies is the association between rs924607 in CEP72 and VIPN. CEP72 encodes for a centrosomal protein that is required for adequate chromosome segregation [63,64]. Centrosomes enable correct alignment of chromosomes during mitosis by controlling the position and orientation of the microtubule spindles at the spindle poles [63,64]. A recent meta-analysis on the effect of this SNP in CEP72 on VIPN in children with ALL confirmed this finding across three studies in the continuation phase of treatment [65]. A clinical trial enrolling newly diagnosed children with ALL and lymphomas is randomizing patients with the high risk (TT) genotype between a decreased dosage (1.0 mg/m^2^) and conventional dosage (1.5 mg/m^2^) of vincristine during the continuation phase of treatment [66]. Recruitment is still ongoing. However, it is important to note that the effect of CEP72 on VIPN likely differs depending on treatment phase and genetic background, since this association was not significant in a white Spanish population in the induction phase and a white Arab population during induction and continuation phases [10,50].

In addition, we assessed the effect of CYP3A5 expression status on VIPN in a meta-analysis and found an overall pooled effect of 0.69 (95% CI 0.38−1.26) (Figure 2). Two studies reported a significant effect of CYP3A5 expression status on VIPN. Renbarger et al. found the strongest association, but it is important to note that they used race as a surrogate for CYP3A5 expression status [14]. This can be debated, since white children can express a CYP3A5 as well, albeit less often than in black children (10−20% and >55%, respectively) [67,68]. Furthermore, Kishi et al. found a significant association in a relatively large cohort (240 children) with a prevalence of 12.5% of severe VIPN [13]. However, all other studies could not replicate these findings, even those with sample sizes adequately powered to detect a difference, such as the population wide GWAS or EWAS. Nonetheless, no study reported CYP3A5 expression as a risk factor for VIPN. In conclusion, this meta-analysis shows that there is no significant effect of CYP3A5 expression status on VIPN.

The comparison and interpretation of the results of the included studies is limited due to heterogeneity in the study population, treatment protocol and varying assessment methods and definitions of VIPN. Firstly, cumulative vincristine dosage varied between 6.5 and 97.5 mg across studies. Since cumulative vincristine dosage likely is an independent risk factor for VIPN, it could be of influence when establishing a relationship between a pharmacogenomic parameter and VIPN [1,3]. However, except for Diouf et al., studies that included cumulative vincristine dosage or treatment protocol as covariates did not find an effect on outcome [8,9,11,13,49,51,52]. Another source of heterogeneity was the variety of measurement approaches to report VIPN. The majority of studies used the NCI-CTCAE for peripheral neuropathy, but other studies used the modified Balis scale, WHO scale, or other methods, to quantify VIPN. The sensitivity and specificity differ across assessment methods and their results can thus not be compared one-on-one [1,3,17]. Similarly, seven studies assessed VIPN prospectively, whereas the other studies assessed VIPN retrospectively. Retrospective VIPN assessment is less sensitive, especially when quantifying the presence of any grade or low grade VIPN [1,3,17]. Moreover, the majority of included studies had a study population of less than 150 patients and were thus limited by a relatively small sample size. This is further reinforced by the observation that most studies assessed the relationship between several genetic variations and VIPN and thus performed multiple comparisons. Therefore, it is advised to counteract the likelihood of false positives by adjusting the significance threshold with for example Bonferroni or False Discovery Rate (FDR) correction [69,70]. This systematic review shows that approximately half of the reported significant associations either did not pass a stringent significance threshold, or that it was not applied. However, associations that did not pass the stringent significance threshold could still be biologically relevant since (stringent) statistical significance should be interpreted within the context of the study design [35,39]. A lack of statistical significance could be an indication that the study was not adequately powered to detect a difference [35,39].

Independent replication is essential to validate the clinical significance of reported associations [33,34]. This could be facilitated by the uniform and reliable assessment of VIPN while employing a sensitive assessment method such as the ped-mTNS or ped-TNV [2,3]. A larger number of patients could be included if VIPN was consistently noted in patient charts. Subsequently, this would allow for reliable comparisons between studies. Furthermore, the growing availability of high-throughput techniques allows for genome- or exome wide analysis in an increasing number of studies. Interestingly, in this systematic review and meta-analysis, the majority of included studies followed a candidate gene approach, limiting the findings to pre-defined selection of genes. The studies that employed population-wide GWA or EWA analysis expanded our knowledge by uncovering genotype-phenotype associations that were not previously described in relation to VIPN. To actualize the potential of pharmacogenomic testing, future studies should apply sensitive and uniform measurement approaches to report VIPN while employing robust genotyping methods such as EWAS or GWAS. These data could be used by consortiums such as the Clinical Pharmacogenetics Implementation Consortium (CPIC) to create guidelines for clinicians to implement pharmacogenomic testing into clinical practice. The CPIC assesses pharmacogenomic studies by assigning levels of evidence and syntheses the results [71]. Another promising approach to assess the influence of pharmacogenomics on VIPN is combining the effect of different SNPs or genetic variations in one effect size, since single SNPs most likely have limited clinical relevance. Such an approach was adopted by Abaji et al., in which a combined-effect model was established to assess the additive effect of several SNPs associated with VIPN [52]. Patients were classified in risk groups according to their weighted genetic risk score; and this model could successfully predict the risk of VIPN in both their discovery and replication cohort [52].

The strength of this systematic review and meta-analysis is the inclusion of all studies assessing the relationship between pharmacogenomic parameters and VIPN. No restrictions were applied regarding patient or study characteristics. Furthermore, we followed the PRISMA guidelines and two independent authors performed screening, data extraction and risk of bias assessment. A standardized risk of bias assessment was performed with the validated EPHPP tool [45]. The weakness of this study is that the meta-analysis was limited to CYP3A5 expression status and that the other pharmacogenomic parameters could not be assessed in a meta-analysis.

## 5. Conclusions

From this systematic review, we can conclude that the following pharmacogenomic parameters have a significant influence on VIPN in children with cancer: SNPs in ABCB1, ABCC1, ABCC2, CYP3A4, GSTM1, VDR, ACTG1, CAPG, CEP72, MAPT, SYNE2, TUBB2B, SLC5A7, BAHD1, COCH, chromosome 12/chemerin, ETAA1, MRPL4, MTNR1B, NDUFAF6, TMEM215 and in three miRNAs. Our meta-analysis shows that CYP3A5 expression does not result in a heightened susceptibility of VIPN. To actualize the potential of pharmacogenomic testing, future research should prospectively assess VIPN with a sensitive measurement tool in both a discovery and replication cohort. Ultimately, the goal would be to develop an individualized protocol based on a patients’ genotype, taking all risk and protective genes into account, and subsequently give patients a dosage that limits the risk of VIPN while maintaining highest possible therapeutic efficacy. Dosage reductions or cessation of treatment, or for some patients even standardized dose capping, would no longer be necessary.

## Figures and Tables

**Figure 1 cancers-14-00612-f001:**
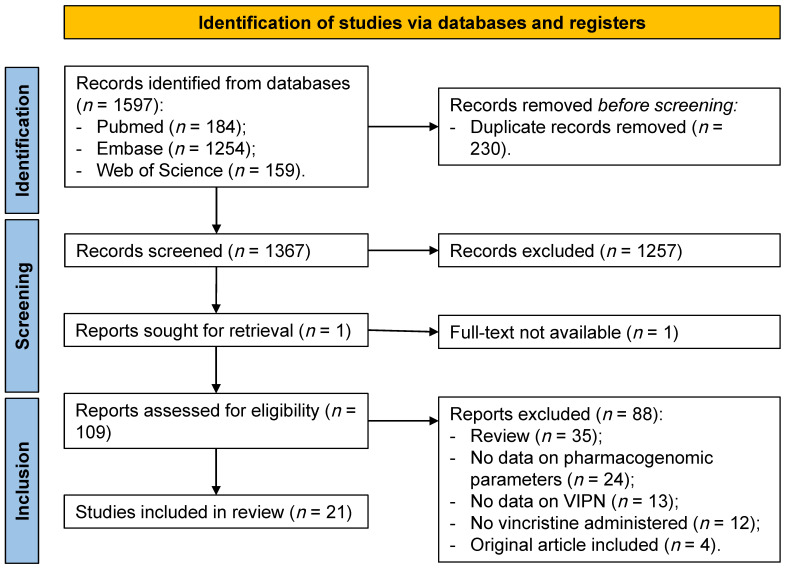
PRISMA flow diagram identification of studies included in the systematic review. VIPN = vincristine-induced peripheral neuropathy.

**Figure 2 cancers-14-00612-f002:**
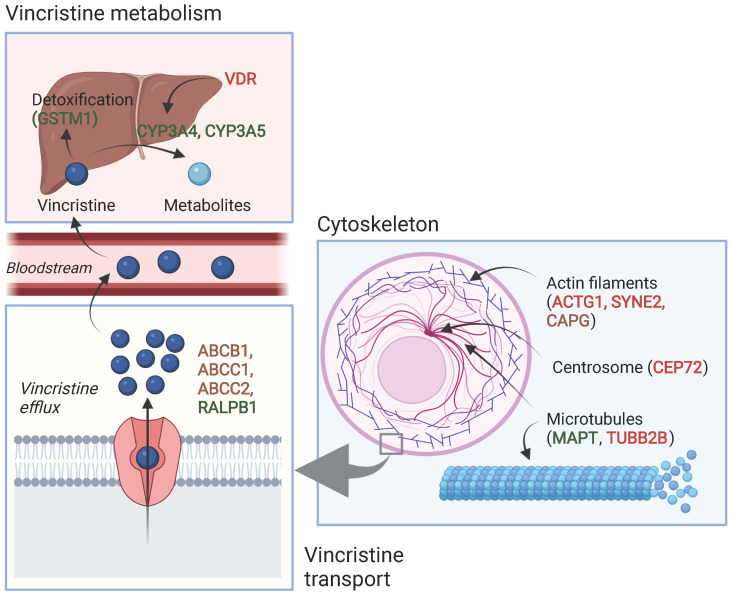
Schematic overview of the function of genes associated with VIPN. Red: described SNPs in this gene are associated with a higher risk of VIPN; green: described SNPs in this gene are associated with a lower risk of VIPN, brown: described SNPs in this gene are associated with both a higher and lower risk of VIPN (different per SNP). Created with BioRender.com.

**Figure 3 cancers-14-00612-f003:**
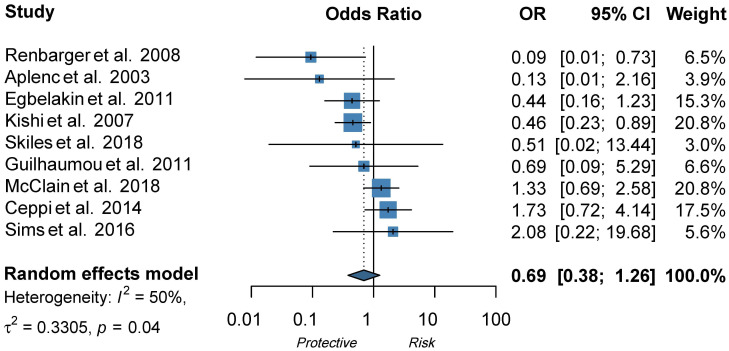
Forest plot showing the effect of CYP3A5 expression status on VIPN, ORs describe the effect of expression of CYP3A5 in comparison to non-expression of CYP3A5 [8,13,14,16,20,28,29,31,32]. The functional allele is *1 and variant alleles are *3 (rs776746), *6 (rs10264272), and *7 (rs41303343). A dominant model was adopted: patients with at least one *1 allele are considered to be expressers of CYP3A5. Patients without *1 allele are considered to be non-expressers of CYP3A5.

## Data Availability

The data presented in this study are available in this article and in the Appendix A.

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
