# Peer review of "Pharmacogenomics of Vincristine-Induced Peripheral Neuropathy in Children with Cancer: A Systematic Review and Meta-Analysis"

_cancers, 2022, doi:10.3390/cancers14030612_

Round 1

Reviewer 1 Report

This is a well-written, precise, correct paper about the vincristine induced neuropathy in childhood cancers. This paper is a negative study on that term, that overviewing the findings of the medical scientific world applying a special approach, there is not any real well-usable conclusion for the clinician. The authors find that there is scarce of strong evidence about the importance of the investigated gene alteration in case of VIPN.

Minor comment: -

Author Response

Thank you for your positive review. We appreciate your time and effort in reviewing our manuscript.

Reviewer 2 Report

Dear Authors,

Thank you for your contribution. I read your article about the pharmacogenomics of vincristine-induced peripheral neuropathy (VIPN) in children with cancer. I think it is a relevant topic in the pediatric field where vincristine is still now a common chemotherapeutic agent. The aim of your article to describe the influence of pharmacogenomic parameters on the development of VIPN is quite useful considering the actual goal to develop individualized therapy based on patent’s characteristics and genotype. However, there is no clear conclusion on the clinical significance of the reported associations between neurotoxicity due to vincristine and single-nucleotide polymorphisms (SNPs) and on the feasibility to perform genotyping in a real word setting.

My major comments after reviewing are the following:

- Introduction: line 49-50, I think it will be better to detail in which types of cancer and in which protocols vincristine is used

-Materials and Methods: paragraph eligibility criteria: what about children with peripheral neuropathy due to other causes (e.g., diabetes, central nervous system malignancy, vitamin deficiency, hereditary causes, nerve compression injury)? Paragraph risk of bias assessment: it is not clear which role had the score you gave to each study (strong, moderate or weak) in your analysis.

- Results: I would specify in the text (not only in the table 1) the methods used for VIPN assessment. It appears in the discussion line 188, although I would give more details on this topic in the results.

My minor comment is the following:

- the tables could be clearer with inside horizontal lines

- I would move table to the end of the article

Kind regards

Author Response

First of all, thank you for reading our manuscript carefully and providing us with valuable feedback that will improve the scientific quality of the manuscript. 

Our point-by-point response to your questions and issues raised can be found in the attachment.  

Reviewer 3 Report

The paper is well written and describes results of a careful analysis.

It may be published as it is

Author Response

Thank you for reviewing our manuscript and we appreciate your positive feedback.